# Receptors Involved in COVID-19-Related Anosmia: An Update on the Pathophysiology and the Mechanistic Aspects

**DOI:** 10.3390/ijms25158527

**Published:** 2024-08-05

**Authors:** Noor N. Al-Saigh, Amani A. Harb, Shtaywy Abdalla

**Affiliations:** 1Department of Basic Medical Sciences, Faculty of Medicine, Ibn Sina University for Medical Sciences, Amman 16197, Jordan; noorsaigh@isums.edu.jo; 2Department of Basic Sciences, Faculty of Arts and Sciences, Al-Ahliyya Amman University, Amman 19111, Jordan; a.harb@ammanu.edu.jo; 3Department of Biological Sciences, School of Science, The University of Jordan, Amman 11942, Jordan

**Keywords:** ACE2 receptor, anosmia, basigin, COVID-19, neuropilin 1, sustentacular cells, TRPV1

## Abstract

Olfactory perception is an important physiological function for human well-being and health. Loss of olfaction, or anosmia, caused by viral infections such as severe acute respiratory syndrome coronavirus 2 (SARS-CoV-2), has received considerable attention, especially in persistent cases that take a long time to recover. This review discusses the integration of different components of the olfactory epithelium to serve as a structural and functional unit and explores how they are affected during viral infections, leading to the development of olfactory dysfunction. The review mainly focused on the role of receptors mediating the disruption of olfactory signal transduction pathways such as angiotensin converting enzyme 2 (ACE2), transmembrane protease serine type 2 (TMPRSS2), neuropilin 1 (NRP1), basigin (CD147), olfactory, transient receptor potential vanilloid 1 (TRPV1), purinergic, and interferon gamma receptors. Furthermore, the compromised function of the epithelial sodium channel (ENaC) induced by SARS-CoV-2 infection and its contribution to olfactory dysfunction are also discussed. Collectively, this review provides fundamental information about the many types of receptors that may modulate olfaction and participate in olfactory dysfunction. It will help to understand the underlying pathophysiology of virus-induced anosmia, which may help in finding and designing effective therapies targeting molecules involved in viral invasion and olfaction. To the best of our knowledge, this is the only review that covered all the receptors potentially involved in, or mediating, the disruption of olfactory signal transduction pathways during COVID-19 infection. This wide and complex spectrum of receptors that mediates the pathophysiology of olfactory dysfunction reflects the many ways in which anosmia can be therapeutically managed.

## 1. Introduction

SARS-CoV-2 is closely related to the causal agents of the pandemic severe acute respiratory syndrome coronavirus (SARS) and the Middle East respiratory syndrome (MERS) and to endemic viruses associated with mild upper respiratory infection syndromes. Brann et al. (2020) reported that infection with SARS-CoV-2 is associated with a high rate of disturbances in smell and taste perception [1]. This olfactory disturbance had a prevalence of over 47% during the first waves of infection, although it was less prevalent with the omicron virus, ranging from 5.8% to 16.7% [2,3]. and was even less according to a meta-analysis of 626,035 patients infected with the omicron virus, ranging between 1.9 and 4.9% from non-European countries and 11.7% from patients of European ancestry, with an overall global prevalence of about 3.7% [4].

The inability to smell (anosmia) and the decreased ability to smell (hyposmia) are more common in the human population than expected, with estimations of incidence ranging widely between 3 and 20% for people older than 70 years [5]. A more recent meta-analysis of a larger sample (175,073 people) reported a 22.2% prevalence in the normal population [6]. This wide range depends on the age, on whether the assessment was done in a subjective or objective setting, and on whether an expanded or brief test was performed, but other risk factors such as viral infections or upper respiratory infections, head trauma, and neurodegenerative diseases are also involved. The problem of anosmia may not seem significant to many patients simply because many patients may not recognize that they have an olfactory disorder and because the olfactory system has significant “plasticity”, represented by a strong interrelationship between the olfactory system and the trigeminal sensory system, creating a uniform flavor experience that may persist after the loss of smell sense [7].

The crosstalk between the olfactory and gustatory sensory systems is thought to occur at the level of the insular cortex. However, the presence of olfactory signal transduction molecules and olfactory receptors in the fungiform taste papilla cells was documented [8].

Olfactory sense is important for the human being not only to localize and appreciate food but also to smell dangerous substances such as gas, fire, or even rotten food. In spite of this significance, many patients with smell disorders’ reported they were not treated properly in their clinics, visiting at least two or more clinics before finding reasonable health care. Landis et al. (2009) reported that 60% of the patients surveyed received unclear or unsatisfactory information about their olfactory problem and its consequences for their health [9]. 

Respiratory viral infections induce an alteration in smell known as post-viral olfactory dysfunction (PVOD), which is usually temporary (short-term) but in some cases may be persistent (long-term). The recovery process for some patients takes up to 2 years [10]. Anosmia is one of the symptoms caused by SARS-CoV-2, which was typically noticeable around the beginning of the infection and was similar to that observed in most other PVODs [11]. It is worth mentioning, however, that a comparison of smell and taste disorder rates in unvaccinated patients with COVID-19 and influenza was made and revealed a higher prevalence of smell and taste disorders in patients with COVID-19 when compared to those with influenza [12]. McWilliams et al. (2022) surveyed two years of follow-up on recovery from COVID-19-induced anosmia and showed that among the 267 respondents, 38.2% reported complete recovery, 54.3% reported partial recovery, and 7.5% reported no recovery at all. Additionally, complete recovery of smell sensation has been observed to be age-related. It was significantly higher in those under 40 years old (45.6% compared to 32.9% in those over 40) [13]. Recent attention has focused on the long-term recovery of smell and taste, alongside the application of objective olfactory assessments. In a prospective multicenter study in Europe, Lechien et al. (2021) documented an objective recovery rate of 85% at 60 days and 95% at 6 months following the onset of olfactory dysfunction [14]. In a recent meta-analysis involving 18 studies involving 3699 patients, it was determined that persistent self-reported olfactory and gustatory disorders at 180 days were 4.3% and 2%, respectively [15]. However, there is currently a lack of long-term (>6 months) outcome data on COVID-19-related olfactory dysfunction in Asians [16].

Moreover, olfactory quantitative dysfunctions (anosmia and hyposmia), as well as qualitative dysfunctions like parosmia (substances that used to smell pleasant now smell foul) and phantosmia (detection of smells without a stimulus), have been reported in patients infected with SARS-CoV-2 variants [17]. It has been reported that vaccinated people infected with the omicron variant of SARS-CoV-2 had symptoms that were shorter and milder than those infected with the delta variant. Anosmia was less common in people infected during the omicron wave than during the delta wave [18].

Acute complete loss or impairment of smell sensation caused by acute infection with SARS-CoV-2 was unfamiliar to physicians and prompted urgency to understand the underlying pathophysiology of this phenomenon. An early suggested mechanism was that SARS-CoV-2 induces damage to the olfactory epithelium in a manner similar to that induced by other viruses such as rhinovirus, coronavirus, parainfluenza virus, or Epstein-Barr virus [10]. This mechanism, as well as many other mechanisms, were proposed and challenged, as discussed in some detail in Section 2 below.

## 2. Hypotheses Explaining Anosmia

Although temporary and persistent olfactory dysfunctions have been observed during the SARS-CoV-2 infection, the etiology of both conditions remains unclear. It was initially hypothesized that coronaviruses, and may be other viruses, caused transient loss or impairment in the olfactory neurons function mainly due to host immune responses such as inflammation caused by cytokines or downregulation of gene expression of odorant receptors and signaling molecules [19,20]. It was proposed that injury to the peripheral and central nervous systems, including olfactory cells, occurs as a result of the massive activation of cytokines [21]. Acute anosmia was found to be correlated with the duration of the SARS-CoV-2 infection and the associated inflammation in the olfactory neuro-epithelium, which may explain the prolonged anosmia in some COVID-19 patients. Experimentally, it was shown that complete olfactory dysfunction induced by SARS-CoV-2 in golden Syrian hamsters lasted as long as the virus persisted in the olfactory epithelium and the olfactory bulb. Indeed, virus and inflammatory mediators’ transcripts were detected in the olfactory mucosa of patients who demonstrated long-term persistence of COVID-19-associated anosmia [22]. In clinical trials, olfactory training and anti-neuroinflammatory therapy achieved positive results in many patients, although 15% of the treated patients did not achieve full recovery of the normal olfactory threshold and almost 5% had no recovery at all [23].

Another hypothesis suggested that olfactory dysfunction resulted from the elimination of the support cells (i.e., sustentacular and Bowman’s gland cells) and the deleterious consequences of this elimination on neuronal cells, such as a decrease in mucus covering the epithelium, a decrease in glucose needed for the function of the cilia, and deciliation or retraction of the olfactory cilia [24]. Moreover, genes encoding uridine diphosphate glucosyltransferases (like *UGT2A1* and *UGT2A2*) were found to differ in different ethnicities, and that was taken as an indication of the involvement of these genes in the variable degrees of anosmia observed during virus infection in different ethnic groups. These enzymes metabolize odorants by glucouronidation in order to clear them from the mucosa, preventing their continuous stimulation of the olfactory receptors (i.e., their saturation) and maintaining the high sensitivity of those receptors [25]. Furthermore, the time of onset of anosmia was of great importance, created some debate, and was taken as a factor pointing to the possible mechanism causing anosmia [26,27]. It was argued that since anosmia occurs rapidly and may recover within a week, this may rule out the probability that anosmia is due to the loss of olfactory neurons, which require minimally 2–3 weeks to regenerate and resume their function [27].

Although several potential mechanisms for COVID-19-induced anosmia have been discussed, the pathophysiological pathways, including the receptors and the intracellular signal transduction molecules involved in olfaction and affected by viral infection, have not been completely elucidated. Emphasizing the mechanisms of virus-induced anosmia may provide therapeutic strategies, especially for patients with persistent post-COVID syndrome. Accordingly, this review sheds light on the possible role of various types of receptors expressed on the olfactory epithelium and their potential roles in signal transduction in anosmia. 

In this review, we focus on anosmia associated with viral infections like SARS-CoV-2, and in particular, we review the literature available on the receptors involved in the smell sense and on the pathways that signals take in order to perceive the smell of an odorant in the brain. The review describes the recent advances that emerged from molecular, physiological, genetic, and imaging studies and highlights many of the remaining questions about post-viral olfactory disturbance. We examine the entry pathways of SARS-CoV-2 through sustentacular cells of the olfactory epithelium and connect this to the mechanism of how it disrupts the action potential generation in olfactory sensory neurons. We bring into focus the significance of ion signaling in one of the pathways by which sustentacular cells and olfactory sensory neurons are affecting each other to detect odorants and how their functionality becomes adversely affected in response to SARS-CoV-2 infection-induced anosmia. 

## 3. The Organization of the Olfactory Epithelium

In mammals, the olfactory epithelium (OE) is located in the most dorsal part of the nasal cavity. OE and the underlying lamina propria, or connective tissue, form the olfactory mucosa [28]. There are three principal cell types in the olfactory epithelium: The olfactory sensory neurons (OSNs), also called olfactory receptor neurons, are either immature or mature olfactory sensory neurons (Figure 1) [29]. OSNs are in direct contact with the inhaled odorants through their cilia that are embedded in the mucus in which odorants dissolve; each OSN extends its dendrite that forms multiple, long, specialized cilia to get in contact with the outer environment, detecting odorants, pathogens, and infectious agents such as respiratory viruses. On the other hand, axons from OSNs in the OE project to the olfactory bulb (OB), where they merge into glomeruli [30] and make synapses with mitral and tufted cells that project to form the olfactory tract. Continuously exposed to the respiratory air that contains pathogens and harmful substances, OSNs have a limited life span of 1–2 months and are continually replaced by newborn neurons generated from the basal cells [31]. 

The second cell type is the supporting sustentacular cells (SCs) and the microvillar cells (MVCs), which together form the brush border. The dendrites of OSNs are surrounded by SCs, whose nuclei and cell bodies line the external layer of the thick neuro-epithelium that is exposed to the outer environment [32,33]. These cell types ensheathe the OSNs and provide them with metabolic support and physical protection. This metabolic support represented by glucose secretion is necessary for the function of the OSN cilia; when these cilia are deprived of glucose, OSNs deciliate and possibly lose their ability to detect odorants [27]. Several roles have been assigned to SC cells, including secretion, endocytosis, detoxification, communication with the basal cells [34], and involvement in the proliferation of the OSNs through purinergic signaling (see Section 8 on purinergic receptors below) [35]. The role of SCs may even be more important than previously thought since they modulate the olfactory thresholds for food odors [36]. In addition, animal experiments showed that SC loss may lead to architectural damage to the OE [37]. Sustentacular cells function as epithelial-like cells when they are involved in the secretion, endocytosis, and metabolism of toxicants, but as glia-like cells when they physically and chemically insulate OSNs. They actively phagocytose dead cells, regulate the extracellular ionic environment [29,34], and play a role in the intercellular signaling mechanisms [38]. Thus, sustentacular cells represent a potential entry door for SARS-CoV-2 into the neuronal sensory system, which is in direct connection with the brain. It is also important to note that sustentacular cells and mature and immature OSNs express gap junction subunits (connexin 43). Zhang et al. used immunohistochemical studies, in situ hybridization, and Western blot analysis to determine the pattern of expression of connexin 43 mRNA in mice’s olfactory epithelium and to verify the presence of connexin protein in nasal tissue. They hypothesized that gap junctions could participate in ion homeostasis in the interstitial fluid that bathes OSNs and sustentacular cells. Furthermore, gap junctions have an important role in the removal of extracellular potassium during periods of high activity of the OSNs. OSNs are electrically coupled to one another or to sustentacular cells at resting membrane potential and uncoupled during odor stimulation, due either to the closure of gap junctions by cAMP or the voltage-dependence of their conductance [39]. Gap junctions would then lower the noise under resting conditions, similar to the role played by cone-cone coupling in visual information processing [40]. These results revealed the role of gap junctions in information processing in the olfactory system since they modulate the membrane properties of mature OSNs [39] and therefore may have a role in the pathophysiological mechanism of anosmia induced by SARS-CoV-2.

The third cell type is the basal cells, which constitute the olfactory stem cells (OSCs), which include horizontal basal cells (HBCs) and globose basal cells (GBCs). HBCs are quiescent under normal conditions, but upon injury, they start dividing vigorously to give the mitotic GBCs and all the other mature cell types of OE [41]. These stem cells are involved in the renewal of the other cell types of OE upon injury or lesions, but they are not known to have any direct role in odorant detection. An interaction seems to occur between HBCs and SCs; direct loss of SCs was found to activate HBCs, which in turn proliferate to substitute for the lost cells and restore olfactory epithelium homeostasis [29]. Additional cell types include Bowman’s gland cells (BGCs) and the olfactory ensheathing glia. The mucus-secreting Bowman’s gland cells also play important roles in maintaining OE homeostasis and function since they produce not only the mucus that keeps the surface of OE moist and prevents its dryness but also secrete certain proteins that aid in the transport of odorant materials to OSNs [1,28,42,43]. Moreover, BGCs transport glucose from the blood vessel through their basal membrane using a glucose transporter and secrete it at their apical membrane into the mucus to nourish the olfactory cilia. Furthermore, these cells, like the sustentacular cells, express two of the viral entry proteins, namely ACE2 and TMPRSS2, thus potentially forming targets for viral entry, which may damage these cells in animals and humans [24,44].

## 4. ACE2 and TMPRSS2 Are the Main Entry Receptors for SARS-CoV-2

The ACE2 receptor has been characterized as the main receptor for the entrance of SARS-CoV-2 [28,45,46], as it was previously reported to be the receptor for SARS-CoV-1 and other human respiratory-related coronaviruses [47]. In airway epithelia, viral uptake is further facilitated by a priming protease called transmembrane protease serine type 2 (TMPRSS2) [48,49]. Cells with high ACE2 and TMPRSS2 expression have strong virus binding capacity and are particularly susceptible to infection (Figure 2) [45,46,50].

Evidence for the role of TMPRSS2 in viral entry was provided by Letko et al. (2020), who demonstrated that the addition of TMPRSS2 during the course of SARS-CoV infection facilitated entry into cells that exhibit low expression of ACE2 receptors [51], and by Chupp et al. (2022), who found that treatment of patients with COVID-19 with camostat methylate, the inhibitor for TMPRSS2, reduced olfactory dysfunction [52]. Due to the location of SCs on the apical surface of the olfactory epithelium, SARS-CoV-2 may first infect these cells, leading to partial or complete breakdown of the olfactory architecture, resulting in a decline in olfaction. These cells have been demonstrated to express a high level of ACE2 and TMPRSS-2 receptors, whereas marker staining of immature and mature OSN demonstrated that ACE2 was not found in olfactory neurons [3]. The presence of these two receptors and their density are important for the initial viral entry, in addition to the viral load. Furin enzymes or the host cell proteases, such as TMPRSS2, which cleave the S protein into S1 and S2, can also modify SARS-CoV-1 and SARS-CoV-2 and facilitate the infection [53]. This cleavage exposes the CendR motif of S1 to the binding pocket at the b1 subdomain of neuropilin (discussed in the next section below), thus potentiating the viral infectivity [54]. ACE2 receptor expression is distributed in a gradient from the nasal epithelium (very high) to the alveoli (low) [55]. In addition, the *ACE2* gene is expressed in the CNS, lungs, and testis [50]. ACE2 messenger RNA is also predominantly expressed in the bronchi and lung parenchyma and in other organs. Using the human kidney 293T cell line, it was shown that overexpression of ACE2 in vitro efficiently enhanced the replication of SARS-CoV-2, whereas neutralization by ACE2 antibodies inhibited replication of the virus in a dose-dependent manner [56]. Ge et al. (2021) developed effective therapeutics and vaccines based on the fact that the viral spike (S) glycoprotein of SARS-CoV-2 mediates viral entry and recognition of the ACE2 receptor. For example, crystal structural comparisons were used to determine the atomic details of possible engagements of the ACE2 receptor in binding the receptor-binding domain (RBD) of the viral spike glycoprotein. Three neutralizing monoclonal antibodies (mAbs) were isolated from SARS-CoV-2-infected individuals that can recognize RBD. Among these three candidate mAbs, P2C-1F11 was the most likely to exert its antiviral activity through functional mimicry of receptor ACE2 and to provide strong protection against SARS-CoV-2 infection in Ad5-hACE2-sensitized mice [57].

The question that remained to be answered was which of these two receptors was more important for virus infectivity. Hou et al. (2020) investigated the relationship between ACE2 receptor expression and SARS-CoV-2 infection by inoculating primary epithelial cultures from several pulmonary regions. They utilized quantitative comparisons of nasal and bronchial airway epithelia obtained as brush samples simultaneously from the same subjects. qPCR results revealed a significantly higher expression of ACE2, but not TMPRSS2, in the nasal tissues than the bronchial tissues. In contrast, the overall expression of TMPRSS2 mRNA was higher in all respiratory tract regions than ACE2 [58].

**Figure 2 ijms-25-08527-f002:**
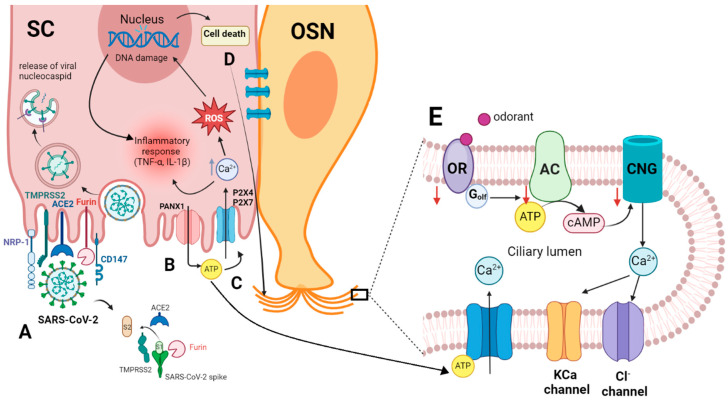
(**A**) Entry of SARS-CoV-2 into the sustentacular cell and the molecular interaction of the virus spike protein and the host cellular receptor ACE2, which depends on spike protein activation by TMPRSS2 and/or furin [46,49,53,59]. (**B**) Viral invasion activates Panx1, promoting the release of ATP from the cell [60]. (**C**) Extracellular ATP binds to P2 receptors, causing an increase in intracellular Ca^2+^ and ROS, which in turn results in DNA damage and cell death. The elevated intracellular Ca^2+^ and DNA damage collectively stimulate inflammatory responses, including the synthesis of cytokines like TNF-α and IL-1ß [61]. (**D**) Intercellular communication of SC with OSN through gap junctions, creating Ca^2+^ waves in the OSN. (**E**) Increasing intracellular Ca^2+^ in OSN activates Ca^2+^-dependent K^+^ channels that may mediate a hyperpolarization and subsequently inhibit the odor response or activate Cl^−^ channels, reducing the negativity in the cell and causing a depolarization. ACE2, angiotensin-converting enzyme 2 receptor; AC, adenylyl cyclase; ATP, adenosine triphosphate; cAMP, cyclic adenosine monophosphate; CD147, basigin; CNG, cyclic nucleotide-gated channel; IL-1ß, interleukin 1 ß; NRP-1, neuropilin-1; OR, olfactory receptor; OSN, olfactory sensory neuron; PANX1, pannexin 1; P2X4, P2X7, purinergic receptors; ROS, reactive oxygen species; SC, sustentacular cell; TMPRSS 2, transmembrane protease serine type 2; TNF-α, tumor necrosis factor-alpha. This figure was created with http://Biorender.com.

## 5. Neuropilin (NRP1) and Basigin (CD147) Are Potential Receptors for Virus Entry

Among the intriguing observations was the ability of SARS-CoV-2 to infect tissues, like the heart or the brain, where the density of ACE2 receptors is low or absent. This observation suggested the presence of another facilitating factor(s) for the entry of the virus. Recent evidence indicates that SARS-CoV-2 can use olfactory neurons and other related regions, such as the olfactory tubercles and para-olfactory gyri, to travel from the periphery into the CNS and may also enter the brain through the blood-brain barrier (BBB). This transit was demonstrated to occur via neuropilin 1 [53,62] or basigin receptors, in addition to cathepsin L (CTSL). In comparison with ACE2 or TMPRSS2, NRP1 and basigin are broadly expressed in the human brain, including the OB [11,63].

Neuropilin-1 (NRP-1) is a one-pass transmembrane receptor that lacks a cytosolic protein kinase domain and is highly expressed in the respiratory and OE. It can enhance the entry of SARS-CoV-2 into the brain through the OE. Functioning as a cell surface receptor, it is unclear whether NRP-1 enables viral attachment or enables receptor-mediated endocytosis in the patient’s cells. However, it was shown that NRP-1 has a b1 domain that binds the CendR binding motif of S1 to potentiate the infectivity of the virus, and this infectivity was blocked by monoclonal antibodies against the b1 binding pocket of NRP-1 [53]. It was postulated that the b1 subdomain of NRP-1 binds with S1 and makes a strong binding with the cell membrane of the host cell, destabilizing the S1/S2 junction, thus bringing quick dissociation of S2 from S1, where S2 then brings membrane fusion of the host and virus and enhances infectivity [64,65]. This interaction of S protein with NRP-1 was reported to be disrupted by natural products like esculetin and 3-methylquercetin, giving these natural products therapeutic potential [66]. It was found that NRP-1 significantly potentiates SARS-CoV-2 infectivity, which was inhibited by a blocking monoclonal antibody against the extracellular b1b2 domain of NRP-1. Analysis of human COVID-19 autopsies revealed SARS-CoV-2-infected NRP-1-positive cells in the olfactory epithelium and bulb, as well as in the endothelial cells of capillaries and medium vessels [62]. Further research measured the gene expression of NRP-1 using single-cell RNA sequencing to examine the detailed expression of NRP-1 in the human brain. Expression of NRP-1 was also assessed by microarray presented as a heat map for six human donors and was detected in the olfactory tubercles and paraolfactory gyri [67].

Likewise, Wang et al. discovered CD147 (basigin or emmprin) as a new receptor for SARS-CoV-2 infection. In cells that were previously reported as resistant to infection with SARS-CoV-2, like BHK-21 cells, the introduction of CD147 alters the virus tropism toward those cells. Moreover, ACE2-deficient T cells can be infected with the SARS-CoV-2 pseudovirus, in which CD147 overexpression facilitates the virus infection. CD147 is a transmembrane glycoprotein of the immunoglobulin superfamily that participates in bacterial and viral infection as well as tumor development and parasitic invasion [59].

The high selectivity of the endothelial cells that form the BBB filters the blood contents available to neurons, and these endothelial cells were found to express high levels of ACE2. But ACE2 receptors are not expressed in the olfactory neurons, whereas SARS-CoV-2 was reported to be located in the olfactory neurons and other brain parts [62]. In order to explain the entry of SARS-CoV-2 into olfactory neurons, an alternative pathway for viral entry was suggested. This alternative pathway was NRP1 receptors, which are expressed in every cell type in the nasal passages, in the olfactory neurons, as well as in the olfactory tract neurons [54]. This solution is acceptable only if it can be proven that the replicating virus (and not the viral particles) is located in the olfactory neurons and other brain areas in significant quantity. This issue is a subject of debate involving proponents and opponents of this suggested mechanism of virus transport from the nose to the brain. The interested reader would benefit from thorough reviews [68,69,70] that challenged different aspects of the alternative pathway, including the methodology (mostly fluorescent microscopy and electron microscopy) and timing of the arrival of the virus to certain brain regions (e.g., hypothalamus), and offered alternative pathways such as the cerebrospinal fluid and the nervus terminalis.

## 6. Odorant Olfactory Receptors Downregulation May Contribute to Olfactory Dysfunction

Odorant olfactory receptors (ORs, odorant receptor proteins) are chemoreceptors expressed in the ciliary membrane of OSNs in the olfactory epithelium [71]. They are G protein-coupled receptors (GPCRs) that are composed of seven transmembrane domains. Each OR can detect more than one odorant due to its allosteric recognition. Odorant perception takes place through these ORs on different neurons, inducing intracellular transduction cascades that send signals to the brain via the olfactory bulb [72].

The SARS-CoV-2 infection of chemosensory cells and the loss of smell are not completely understood. One of the possible pathophysiological mechanisms of smell loss caused by the SARS-CoV-2 virus in COVID-19 patients was suggested to be the downregulation of olfactory receptor proteins within the olfactory sensory neurons [11,20]. This assumption was supported by a recent study that showed a significant downregulation of olfactory receptor genes in two types of chemosensory cells: olfactory and taste cells infected with the human coronaviruses HCoV-OC43 and SARS-CoV-2. Among the ORs reported to be downregulated are OR51E1, OR7D4, and TAAR1 (trace amine associated receptor 1) [21]. Similar observations have been reported by Verma et al. (2022), who showed that SARS-CoV-2 infection of sustentacular cells of experimentally infected animals caused inflammation, extensive ciliary damage, and downregulation of OR expression in OSNs, leading to olfactory dysfunction [19]. In a mouse model, OR genes as well as other molecules involved in olfactory signal transduction were found to be downregulated [73], although the same was not found in human patients [74]. The downregulated proteins include G_olf_, CNGA2, ACIII, and ADCY3, another odorant-related signaling molecule, which are all important molecules in olfactory signal transduction [19]. Whether the downregulation of these OR occurs as a result of the infection (i.e., after the infection) or is a cause of the anosmia remains to be resolved [20,27].

Another possible mechanism for anosmia associated with viral infection is the induction of apoptosis in olfactory sensory neurons during the early stage of infection. Such mechanisms may provide a defensive mechanism that prevents the invasion of the CNS from the periphery. This was concluded from a study conducted on female C57BL/6 mice infected with influenza A virus in the nostrils. Using immunohistochemistry and dual immunofluorescent labeling techniques, it was revealed that the expression of Fas ligand molecules was upregulated and the c-Jun N-terminal kinase (JNK/c) signal transduction pathway, which is involved in the process of apoptosis, was activated in virus-infected OSNs. Furthermore, Iba1-expressing activated microglia/macrophages have a role in phagocytic activities and help clear apoptotic bodies [75]. But as we pointed out in the Section 2 hypotheses explaining anosmia above, the time course of the regeneration of the OSNs does not correspond to that of the recovery of smell sense. Recovery of smell sense takes place in about 5–7 days, whereas regeneration of OSN after apoptosis may take 2–3 weeks [27], suggesting that this apoptosis may serve as a feedforward mechanism that prevents the spread of the virus to the brain. Overall, in animal experiments but not in humans [74], OR genes, other proteins, and odorant molecules that are responsible for detecting odorants and initiating signal transduction to the brain are downregulated during infection [73]. SARS-CoV-2 infection may cause smell loss by downregulating the olfactory receptor and inducing apoptosis in olfactory sensory neurons. The activation of apoptosis in infected neurons may prevent viral spread to the brain, assuming that the spread to the brain is through the olfactory nerve, although this hypothesis is challenged [70].

## 7. Transient Receptor Potential Vanilloid (TRPV1) Trafficking May Lead to Odor Inhibition

TRPV1 is a family of transient receptor potential (TRP) cation channels that are permeable to Ca^2+^ and expressed in many tissues, including the nervous tissue. Six TRPV channel subtypes (TRPV1-6) have been identified [76]. TRPV1-4 are localized in OE. Using immunohistochemistry and double staining techniques, it was found that TRPV1 is expressed in various regions of the OE of CBA/J mice. TRPV1 immunofluorescence was detected in the olfactory cilia, basal cells, sustentacular cells, and OSNs. TRPVs may perform several functions in the OE since they may participate in olfactory adaptation, olfactory/trigeminal interactions in nasal chemoreception, and OE homeostasis. It was also suggested to be involved in olfactory transduction and olfactory dysfunction related to sinonasal inflammatory disease [77].

Olfactory dysfunction during infection may be related to the overactivity of the trigeminal afferent system that stimulates calcitonin gene-related peptide (CGRP) release and inhibits the detection of the odor by the olfactory receptor [78]. TRPV1 was found to be expressed on most of the trigeminal afferents innervating the nasal mucosa. During SARS-CoV-2 infection, it was shown that the virus triggers the inflammatory response, which is accompanied by an increase in the production of cytokines, specifically TNF-α [79], and consequently stimulates the trigeminal system in the nasal mucosa (Figure 3).

TNF-α activates the trigeminal ganglion by enhancing the expression of TRPV1 channels and many cytokines, including IL-1b and brain-derived neurotrophic factor (BDNF). In response to TNF-α and IL-1b stimulation, activated trigeminal ganglion neurons synthesize and release CGRP [80]. TNF-α enhances the expression and trafficking of TRPV1 and TRPA1 to the surface of trigeminal neurons, and such trafficking is required for CGRP exocytosis [81]. Experimentally, it was shown that the CGRP receptor is expressed on the OSNs, and when it binds its ligand, CGRP, it inhibits odor detection by the olfactory receptors. In addition, it was found that cAMP is produced by OSNs in response to CGRP application, presumably through coupling of the CGRP receptor with adenylyl cyclase III [82]. cAMP mediates the inhibitory olfactory transduction pathway by activating a cyclic nucleotide-gated (CNG) channel, allowing Ca^2+^ permeation. Ca^2+^ then activates a Ca^2+^-dependent K^+^ channel (K_Ca_), resulting in a hyperpolarizing receptor potential [83].

In summary, TRPV1 is expressed in the olfactory epithelium and plays a role in olfactory adaptation, nasal chemoreception, and OE homeostasis. TRPV1 mediates the inhibitory odor response during SARS-CoV-2 infection. The virus induces an inflammatory response, increasing TNF-α, enhancing the expression and trafficking of TRPV1 and TRPA1, and stimulating the trigeminal system, which releases CGRP that inhibits odor detection by olfactory receptors.

## 8. Purinergic Receptors Involvement in Olfactory Signaling (P2X and P2Y Receptors)

Purinergic receptors are expressed on the plasma membrane of neurons, basal, and sustentacular supporting cells of the olfactory epithelium and play an important signal transduction role through binding to the extracellular nucleotides [84,85]. Besides the importance of this system in modulating physiological processes like apoptosis, thromboregulation, cell proliferation, platelet aggregation, endothelial vasodilation, and pain, it also maintains the integrity of the olfactory epithelium by inducing proliferation and differentiation of OSNs and regulating olfactory sensation [61].

Purinergic receptors can be initially classified into P1 and P2 groups according to their structural and functional characteristics. P1 receptors are G protein-coupled receptors and are sensitive to a group of signaling molecules, including ATP, ADP, AMP, and adenosine, which are released by all cells as a response to cellular damage caused by the action of pathogens. P1 receptors are subclassified into A1, A2A, A2B, and A3 [86]. On the other hand, ATP, ADP, UTP, and UDP are extracellular nucleotides that bind to a specific subtype of purinergic receptors (P2). P2 is further subdivided into two categories: P2X and P2Y receptors. The P2X receptor is a ligand-gated ion channel that responds rapidly when bound to its key ligand, ATP. It is a nonselective cation channel that allows Na^+^, K^+^, and Ca^2+^ flux. In contrast, the P2Y (1, 2, 4, 6, 11–14) receptor is a G protein-coupled receptor that activates a variety of second messengers and intracellular signaling pathways, thus acting at a slower pace than P2X [31,87].

To detect the localization of P2X receptors in OE, Gao et al. used RT-PCR and immunohistochemistry to monitor the expression of these ionotropic purinergic receptors in adult mouse OE. They targeted type III neuron-specific tubulin (TuJ1) using polyclonal rabbit anti-P2X1-3 and monoclonal mouse anti-TuJ1. Their results showed that P2X1-3 receptors were localized at the basal part of OE, including the sustentacular cells, OSNs, and basal cells. P2X1 and P2X2 were mainly localized to the cell bodies of sustentacular cells and basal cells. Strong co-localization of P2X3 with TuJ1 was observed in the olfactory nerve. An immunoreactivity to P2X4-7 receptors was also detected in the OSNs, and to P2X5 receptors in the axons of OSNs only. The localization of P2X receptors in the basal progenitor cells of adult mouse OE may explain the role of purinergic receptors in regulating adult OSN neurogenesis [85].

Extracellular ATP participates in the regeneration of the sensory neurons after epithelial damage. The P2X receptor is expressed in the basal cells of the olfactory epithelium, and blockade of this receptor inhibited the injury-induced proliferation of basal cells [85]. In addition, it was reported that intranasal ATP and purinergic agonists induced proliferation of the basal cells in the olfactory epithelium, and an opposite effect was observed when purinergic antagonists were used [88]. The mechanism of ATP-induced olfactory epithelium proliferation through purinergic receptor activation was mediated by growth factors like fibroblast growth factor 2 (FGF2), transforming growth factor alpha (TGFα), and neuropeptide Y (NPY) [89]. Injured OSNs usually release ATP, which activates P2X and P2Y receptors and stimulates calcium signaling in microvillar cells and sustentacular cells, thus enhancing the release of NPY. NPY activates Y1 receptors expressed on basal cells, thus stimulating mitogen-activated protein kinase (MAPK), resulting in the regeneration of new OSN [31].

Persistent anosmia appears to be related, in part, to defects in olfactory neurogenesis in response to various pathogenic infections that stimulate neuro-inflammation and oxidative stress. Many studies indicate that SARS-CoV-2 initially causes neuro-inflammation that leads to damage in the OSNs of the nasal neuroepithelium and spreads through the OB to different regions of the brain [90]. SARS-CoV-2 infection was found to increase the level of extracellular ATP, leading to abnormal activation of purinergic receptors, which may cause the hyperactivation of P2X7 receptors that facilitate consequent neuro-inflammatory processes [91,92]. Although activation of both P2Y and P2X receptors by ATP caused a remarkable decrease in odor responsiveness [93], P2X and P2Y receptors exhibited different effects on olfactory transduction [94].

Purinergic receptors, such as P2Y, are GPCRs that may interact with specific subfamilies of olfactory receptors to enhance cell surface expression and modulate odorant responsiveness. The specificity of olfactory receptor (M71) interactions with other GPCRs was examined by Bush et al. (2007). They co-expressed the olfactory receptor (M71) with purinergic receptors (P2Y1R and P2Y2R) in HEK-293 cells and showed that co-expression of the olfactory receptor with purinergic receptors (P2Y1R and P2Y2R) enhanced cell surface expression of the olfactory receptor [95]. Moreover, activation of P2Y receptors in sustentacular cells of mouse olfactory epithelium was mediated by the phosphorylation of the transcription factor cAMP-response element binding protein (CREB), causing odorant detection. This effect was reduced when a purinergic receptor antagonist like suramin was used [38]. Nevertheless, cAMP-dependent protein kinase was found to have a role in SARS-CoV-2 infection in Vero E6 cells. The interaction of CREB and CREB-binding protein (CBP) was found necessary for viral replication since the knockdown of CREB and CPB with siRNA inhibited the viral infection [96]. The conclusion to be drawn is that purinergic receptors, and may be other cAMP-related receptors, are important for assisting olfactory receptors to be localized into the surface membrane and consequently in their odor responsiveness.

CREB is activated in response to extracellular stimuli by phosphorylation at residue Ser-133 [97]. This activation depends on adenylyl cyclase III (ACIII)-mediated olfactory signaling and upon activation of P2Y purinergic receptors on sustentacular cells. Odorant-induced ATP release from OSNs leads to long-term activation of CREB in neighboring sustentacular cells via activation of their P2Y receptors and subsequent phosphorylation of CREB. In support of this observation, the application of exogenous ATP was found to induce CREB phosphorylation in the nuclei of the sustentacular cells. CREB activation was shown to be purinergic receptor-dependent, where the extracellular nucleotides are an essential tool in communication between OSNs and sustentacular cells [35,98].

Interestingly, preincubation of OE with SQ22536, an adenylyl cyclase antagonist, inhibited the activation of CREB in mature and immature OSNs, indicating that ATP-mediated activation of CREB in these cells depends completely on ACIII signaling. In sustentacular cell nuclei, this inhibitor was able to reduce but not abolish the levels of CREB phosphorylation, indicating that ATP-induced phosphorylation of CREB in sustentacular cells is only partially dependent on ACIII signaling [38]. It is worth mentioning that in the previous section on the role of TRPV1 receptors, we reviewed the literature that indicated that cAMP mediates the inhibitory olfactory transduction pathway by activating a cyclic nucleotide-gated (CNG) channel, allowing Ca^2+^ permeation, and activating a Ca^2+^-dependent K^+^ channel (K_Ca_), resulting in a hyperpolarizing receptor potential. This seemed to contradict the role of cAMP/CREB described in this section. This contradiction was reflected in the question raised by Madrid et al. (2005), inquiring how an OSN would generate two opposite responses independently of each other if they shared a common transduction pathway (i.e., AC/cAMP/CREB/pCREB). These authors speculated that the transduction proteins that participate in each response type form complexes somehow segregated from each other within the cilia, thus allowing them to operate as independent functional units [83].

This intercellular communication between olfactory neurons and their supporting counterparts, the sustentacular cells, indicates the importance of sustentacular cells in the development of smell sensation. In fact, it has been shown that as the immature ciliated dendrites of OSN emerge to the epithelial surface, they undergo a sideways migration to become enwrapped within the sustentacular cells during their maturation process [98]. This architectural arrangement could potentially modify the function of OSN since the wrapping sustentacular cells are responsive to many chemicals, including purinergic, cholinergic, and endocannabinoid mediators, thus producing long-lasting potentials that modify OSN activity [98]. Therefore, the odorant-mediated activation of OSNs, which causes them to fire their action potentials, depends on intact sustentacular cells, through the induction of CREB phosphorylation, and on extracellular nucleotides that mediate this intercellular communication [38]. Single-cell RNA sequencing revealed that only sustentacular cells and horizontal stem cells in the olfactory cleft, as well as the vascular pericytes in the olfactory bulb, co-express ACE2 and TMPRSS2, which explains why the local infection of these cells may cause a block of odor conduction, indicating that general disturbance of the mucosal architecture can obstruct the processing and signal transmission to the brain [99]. Invasion of sustentacular cells by SARS-CoV-2 is considered a vital factor in disintegrating the architecture of OE, a matter that has transient detrimental effects on smell perception.

In COVID-19, ATP is released by infected cells through pannexin 1 (PANX1) channels as a response to conditions like viral invasion, cell stress, and hypoxia. Once ATP is released, it remains for hours to days in the extracellular environment, exerting its pro-inflammatory effects by initiating a cascade that activates purinergic receptors until it is metabolized. Due to its chemoattractant ability for macrophages, ATP is recognized as a damage-associated molecular pattern that activates P2X and P2Y purinergic receptors [60].

It was reported that activation of both the P2Y and P2X receptors by exogenous and endogenous ATP caused a remarkable decrease in odor responsiveness [93]. A low level of endogenous ATP was found to increase Ca^2+^ and suppress odor sensitivity, whereas superfusion of receptor antagonists like suramin and pyridoxal-phosphate-6-azophenyl-2′,4′-disulfonic acid (PPADS) reduced the basal Ca^2+^ level and increased sensitivity to odors [93]. High levels of extracellular ATP result in worsening of the inflammatory responses through PANX1/P2 receptor activation and play an essential role in acute pulmonary inflammation, edema, and lung dysfunction after ischemia/reperfusion injury. Damaged cells release ATP, which activates P2X and P2Y purinergic receptors on the neighboring cells of the olfactory epithelium [31]. These data indicate that ATP release in the olfactory epithelium after noxious stimuli may serve a physiological role as a neuroprotective mechanism. Moreover, the anosmia associated with COVID-19 infection may serve as a protective mechanism.

In summary, purinergic receptors on the plasma membrane of neurons and supporting cells in the olfactory epithelium play a key role in signal transduction, cell proliferation, and maintenance of olfactory epithelium integrity. SARS-CoV-2 infection can increase extracellular ATP, leading to an abnormal activation of purinergic receptors, which causes inflammation, neuro-inflammation, and olfactory dysfunction. Activation of these receptors is also crucial for olfactory signal transduction and the proliferation of sensory neurons, but during infection, it may contribute to anosmia as a protective mechanism against viral spread.

## 9. Interferon Gamma and Interleukins Receptors in Sustentacular Cells

Interferon-gamma (IFN-γ) is a soluble cytokine produced mainly by activated T lymphocytes, macrophages, mucosal epithelial cells, and natural killer cells. It modulates antiviral and antibacterial immune responses during infection. IFN-γ mediates cellular responses through its binding to its heterodimeric cell-surface receptor (IFN-γ R), which in turn activates downstream signal transduction pathways and subsequently affects the regulation of gene expression [100]. Lack of IFN-γ receptor was associated with invasion and persistence of influenza virus A/WSN/33 in the mouse olfactory system [101], whereas increased IFN-γ expression was correlated with the immune response to sinonasal bacterial biofilms in surgical chronic rhinosinusitis patients [102]. It has been suggested that IFN-γ mediates the cytokine pathways that may contribute to the pathogenesis of chronic rhinosinusitis-associated anosmia. This assumption has been investigated using electro-olfactography recordings to assess the odorant responses of transgenic mice with enhanced production of IFN-γ by olfactory sustentacular cells. The results showed that chronic IFN-γ expression caused a significant decrease in odorant responsiveness, which was explained by reduced olfactory receptor expression [103]. Although there was no inflammatory tissue damage reported in this case, suggesting a direct effect of IFN-γ, it seems that increased levels of IFN-γ and other cytokines may be involved in COVID-19-induced olfactory disturbance.

As mentioned earlier in this review, inflammation in OE was suggested as one of the possible mechanisms for COVID-19-induced anosmia. Inflammation causes damage in olfactory neurons and activates horizontal basal stem-cell-mediated neurogenesis, but prolonged inflammation arrests those cells in an undifferentiated state by stimulating “stemness”-related transcription factors, leading to olfactory loss. It has been shown that the expression of IFN-γ as well as other cytokines and chemokines was upregulated in the whole olfactory mucosa of the inducible olfactory inflammation mouse model in response to continuous stimulation by tumor necrosis factor (TNF) [104].

A series of physiopathological mechanisms are related to the disease caused by SARS-CoV-2 infection, in which the infection mobilizes a wide variety of biomolecules, mainly proinflammatory cytokines such as IL-1, IL-6, IL-12, IFN-γ, and TNF-α, which preferentially target lung tissue [105]. Interferon signaling has a crucial role in controlling and regulating disease severity after infection with many viruses, including coronaviruses. SARS-CoV-2 was found to be sensitive to type I and III interferons in human cells in vitro and in mice in vivo [106]. These authors infected double-knockout C57BL/6J mice for type I and II interferon receptors and wild-type controls with 10^4^ plaque-forming units of SARS-CoV-2 MA10. Four days post-infection, mice displayed much higher congestion scores, which were associated with higher viral titers on 2 and 4 days post-infection compared to the wild-type controls. These data suggest that IFNs are important in limiting viral replication and assisting in virus clearance in vivo. Consistent with these data, lung function abnormalities were more pronounced and prolonged in infected double-knockout mice [106]. On the other hand, the role of IL-6 is controversial. Cazzolla et al., for example, investigated the role of IL-6 in smell and taste disorders. A venous blood sample from 67 COVID-19 patients was used to measure IL-6 levels using the chemiluminescence assay. A significant correlation was found between IL-6 levels and the type of dysfunction. The olfactory and gustatory dysfunctions had a higher score in patients with higher levels of IL-6. Also, the patients with both disorders had even higher levels of IL-6 [107]. Other studies showed a weak correlation between IL-6 and olfactory dysfunction, but these studies were also complicated by the invalid use of correlation between the parameters studied, by comparison of severe COVID-19 to non-severe cases, or by the correlation of lower IL-6 levels with mild cases of disease (reviewed in 3). This issue may remain controversial and awaits a well-controlled study and/or a meta-analysis to provide a clear picture.

Interferon and interleukin levels are inversely proportional to each other. This unique and inappropriate inflammatory response was observed, and it has been assumed that increased levels of IL-6 in COVID-19 patients would suppress interferon levels [108]. The relation of serum IL-6 and IFN-γ concentrations to mortality in the cohort of patients was investigated. It was found that over 60% of peripheral T cells in severely ill COVID-19 patients were unable to produce measurable IFN-γ when stimulated with the potent IFN-γ mitogen phytohemagglutinin. This defect in IFN-γ production was associated with increased levels of IL-6 [109].

In brief, IFN-γ, a cytokine produced by immune cells, modulates antiviral and antibacterial responses during infection. Increased IFN-γ expression is linked to immune responses and olfactory dysfunctions, such as anosmia, in conditions like chronic rhinosinusitis and COVID-19. In COVID-19 patients, high levels of proinflammatory cytokines, including IL-6, correlate with reduced IFN-γ levels, contributing to severe disease outcomes and olfactory disturbances. The role of IL-6 in COVID-induced anosmia, however, remains unclear, as discussed above.

## 10. Role of Epithelial Sodium Channel (ENaC) in COVID-19 Induced Anosmia

The epithelial sodium channel α-subunit (ENaC-α) is a channel protein that is responsible for maintaining the balance of salt and water in the epithelium of many organs, including the lungs and olfactory epithelium [110,111]. It is also expressed in stem cells as well as in the nervous system, such as in brain centers controlling fluid volume or blood pressure, the retina, and the olfactory bulb [112].

Olfactory dysfunction induced by COVID-19 can be linked to impairment of function [105]. In sustentacular cells, ENaC helps maintain ion gradients in the liquid layer that surrounds the olfactory epithelium [111], a matter that is important to the proper function of OSNs. Impaired ENaC activity in the nasal epithelium is associated with chronic rhinitis, characterized by a continuous runny nose [113].

It has been shown that the co-expression of SARS-CoV-1 envelope, SARS-CoV-2 envelope, and SARS-CoV-2 spike protein with ENaC in *Xenopus* oocytes causes a remarkable inhibition in epithelial sodium channel activity [114]. Furthermore, a study demonstrated that the knockout of ENaC in the olfactory bulb of mice resulted in a decreased number of newly generated neurons and inhibited adult neurogenesis [112]. Therefore, the reduction in ENaC functionality may impact adult neural stem cell proliferation in the olfactory bulb and, consequently, the olfaction process.

In humans, ENaC-α is often found in the same types of lung and respiratory tract cells as ACE2 [110]. Anand et al. (2020) performed systematic expression profiling of ACE2 and ENaC-α across 65 published human and mouse single-cell studies, including ~1.3 million cells, using the nferX single-cell platform. Their analysis showed a significant overlap between the expression of ENaC-α and the viral receptor ACE2. SARS-CoV-2 developed a unique S1/S2 cleavage site that is absent in any previous coronavirus sequence, and this site can be cleaved by furin enzymes. On the other hand, the ENaC α-subunit has an identical furin-cleavable peptide. This shows an astonishing imitation by the virus of the sodium channel α-subunit. This suggests that SARS-CoV-2 may use the same protease (furin) that activates ENaC-α for the purpose of getting inside human respiratory cells [110]. In the case of the COVID-19 infection, this will lead to compromised ENaC activity, compromised fluid reabsorption, and the lung pathology observed in these patients.

On the other hand, Ozdener et al. (2022) used various molecular techniques to show that TRPV1 is co-localized in cultured adult human fungiform taste cells that express the ENaC δ-subunit and that modulating TRPV1 activity by high salt media and capsaicin can alter ENaC mRNA expression. Moreover, renin-angiotensin-aldosterone system (RAAS) components function in a complex along with ENaC and TRPV1 expressed in cultured adult human fungiform taste cells. Changes in ACE2 receptor expression can alter the balance between the two major RAAS pathways, ACE_1_ (Ang II; At_1_R) and ACE_2_ (Ang (1–7); MasR1), leading to changes in ENaC expression and responses to NaCl in salt-sensing human fungiform taste cells [115].

## 11. Physiological Role of Oscillatory Calcium Transients in Sustentacular Cells and Signal Transduction through Olfactory Sensory Neurons

Sustentacular cells express purinergic receptors (P2Y), and they respond to P2R agonists ATP and UTP with an increase in intracellular calcium. This Ca^2+^ will be utilized later, either to develop intercellular or intracellular Ca^2+^ signals. Using confocal calcium imaging of neonatal mouse olfactory epithelium slices, Hegg et al. observed spontaneous intercellular calcium waves and intracellular calcium oscillations in sustentacular cells. This may make sustentacular cells chemically coupled and capable of transferring changes in intracellular calcium between cells through gap junctions [34].

When P2Y is stimulated by ATP, diverse intracellular signaling pathways are activated, including the activation of PLC, PKC, and CaM kinase. PLC activation results in the formation of IP3, causing the release of Ca^2+^ from IP3-sensitive Ca^2+^ stores, thus increasing [Ca^2+^]i [29,34,38]. Some P2Y receptors couple to adenylyl cyclase and increase the production of cAMP [38], with significant consequences as described in the purinergic receptors section above.

Dynamic intracellular calcium fluxes in sustentacular cells can serve many different signals in the olfactory epithelium, including secretion, proliferation, the development of sustentacular cells, and the release of chemical signals via calcium-dependent exocytosis. This transient increase in intracellular calcium signal may be translated to an electrical membrane signal through K^+^ and Na^+^ channels [31]. In mouse sustentacular cells, Ca^2+^ waves could serve as a “housekeeping” mechanism to maintain a functional OE. One of the consequences of the Ca^2+^ increase in these cells is the increased opening of calcium-dependent K channels (K_Ca_) for clearance of K^+^ to achieve the level of hyperpolarization that is necessary to increase the availability of Na^+^ channels and allow sustentacular cells to fire action potentials that may propagate across gap junctions [116]. These observations point to a key role for sustentacular cells in maintaining a stable intra- and intercellular ionic microenvironment in the olfactory epithelium.

In mammals and other vertebrates, once the receptor of OSN binds an odorant molecule, the ligand-bound receptor activates a G protein, and a cascade of events is initiated that activates an adenylyl cyclase (ACIII). The cyclase converts the abundant intracellular molecule ATP into cyclic AMP, which binds to the intracellular aspect of the cyclic nucleotide-gated (CNG) ion channel, enabling it to conduct cations such as Na^+^ and Ca^2+^. Inactive OSNs normally maintain a resting voltage across their plasma membrane of about −65 mV. When the CNG channels open, Na^+^ and Ca^2+^ ions influx, causing the inside of the cell to become less negative. If enough channels are opened long enough, they cause the membrane potential to become about 20 mV less negative; the cell reaches threshold and generates action potentials. The action potentials are then propagated along the axon, which crosses the cribriform bone plate into the forebrain, where it synapses with second-order neurons in the olfactory bulb.

As long as a positive charge enters the CNG channel, it will be able to activate another ion channel that is permeable to the negatively charged chloride ion. Normally, neuronal Cl^−^ channels mediate an inhibitory response, as Cl^−^ ions tend to be distributed in such a way that they will enter the cell through an open channel. Unusually, OSNs have a high intracellular Cl^−^ concentration maintained via a membrane pump to allow a Cl^−^ efflux when these channels are activated. This mechanism will add a net positive charge to the membrane that further depolarizes the cell, thus increasing the excitatory response magnitude. Thus, the OSN maintains its own Cl^−^ voltage in case the Na^+^ gradient in the mucus is insufficient to support a threshold current and uses it to boost the response [117,118], or when the Na^+^ gradient in the mucus has been disturbed, as when ENaC-α function has been compromised when SARS-CoV-2 invades the sustentacular cells.

## 12. Conclusions

Prolonged olfactory impairment after SARS-CoV-2 infection is a serious problem that requires further investigations to understand the pathology of this deterioration. Significant efforts have been made to explore, at the cellular and molecular levels, the mechanisms of the viral invasion and the signal transduction that leads to olfactory dysfunction. In this regard, the role of the many receptors and channels that may be involved in olfactory dysfunction during COVID-19 was discussed. The sustentacular cells seemed to be the starting site for olfactory dysfunction due to the higher density of ACE2 and TMPRSS2 receptors in these cells than in other types of cells in the olfactory epithelium (e.g., the olfactory sensory neurons, which may not have any significant density of those receptors). In addition, the presence of neuropilin-1 in the plasma membrane of SCs seemed to mediate the binding of the CendR motif of the S1 protein and enhance the infectivity of the virus. A similar role has been ascribed to basigin (CD147). Moreover, the inflammatory response caused by the viral invasion and the concomitant release of proinflammatory cytokines (i.e., TNF-α) may enhance the expression and trafficking of TRPV1 receptors, which may help to enhance CGRP synthesis and release, a process that seems to be involved in odor inhibition. Furthermore, viral infection stimulates Panix1 channels, which release ATP extracellularly to bind P2 purinergic receptors that mediate Ca^2+^ release and intercellular communication with OSNs through gap junctions, creating Ca^2+^ waves in these OSNs. These Ca^2+^ waves activate Ca^2+^-dependent K^+^ channels that mediate hyperpolarization. Although ATP has an excitatory role on the Ca^2+^ responses in non-stimulated OSNs, when co-applied with the odorant, it showed an inhibitory role on those responses, indicating its important modulatory role [93]. This wide and complex spectrum of receptors that mediates the pathophysiology of olfactory dysfunction reflects the many ways in which anosmia can be targeted to develop therapeutic approaches.

Additional research is needed to comprehensively understand the molecular processes and the electrophysiological alterations that impact various aspects of olfactory function during viral infections, including the detection threshold, odor discrimination, and odor identification. As new variants of the virus emerge, the occurrence of anosmia may vary, and alterations in their molecular mechanisms will change. Continued investigation is necessary to clarify the pathogenesis and differences in olfactory dysfunction among different variants.

## Figures and Tables

**Figure 1 ijms-25-08527-f001:**
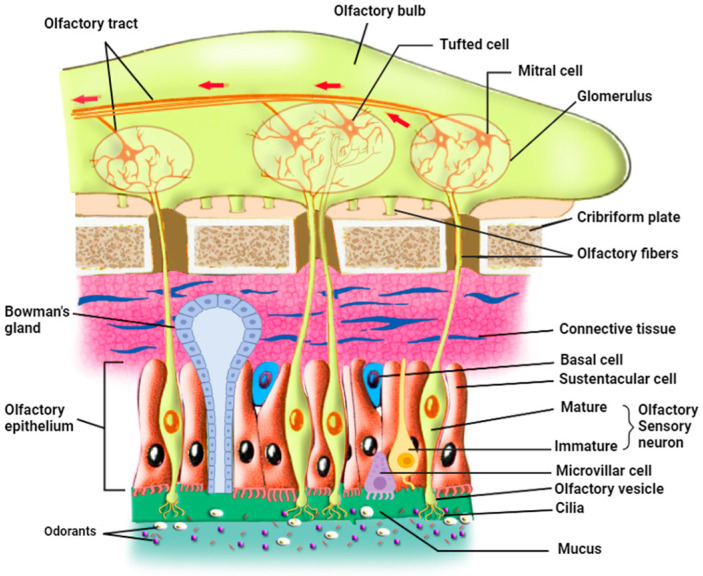
Organization of the olfactory epithelium and its communication with the olfactory bulb (prepared with http://Biorender.com).

**Figure 3 ijms-25-08527-f003:**
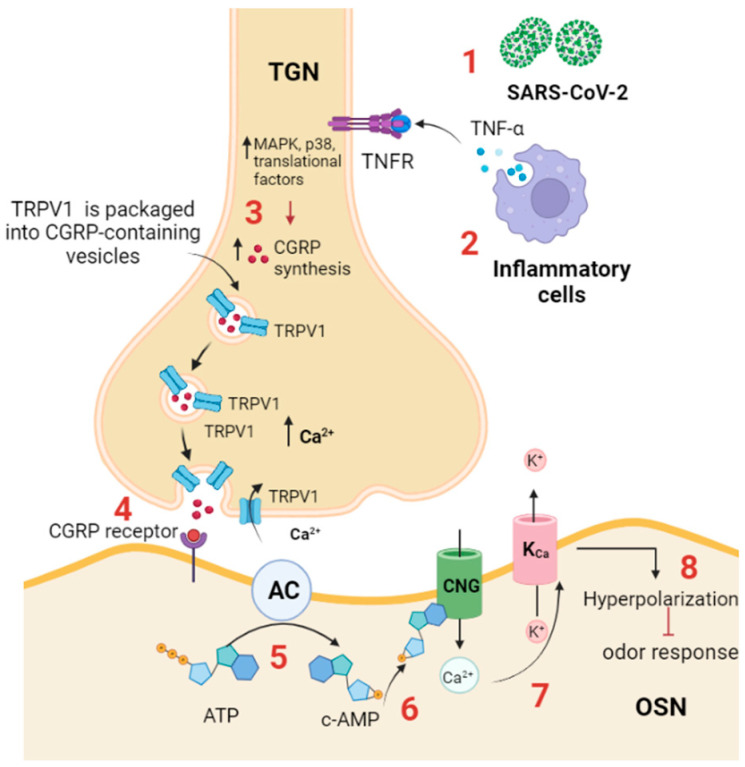
TRPV1 participates in anosmia associated with SARS-CoV-2 infection. (1) SARS-CoV-2 infection activates the immune response; (2) inflammatory cells produce proinflammatory cytokines, such as TNF-α, which binds to its receptor (TNF-αR) expressed on the trigeminal ganglion neuron in the OE; (3) TNF-α activates intracellular cascades, including MAPK, p38, and other translational factors, to increase CGRP synthesis and increases the expression and trafficking of TRPV1 to the plasma membrane, thus triggering CGRP release. (4) CGRP binds to its receptor (CGRPR) expressed on OSN. (5) CGRPR activates adenylyl cyclase, which in turn converts ATP to cAMP; (6) cAMP activates a CNG channel, allowing Ca^2+^ influx; and (7) Ca^2+^ activates a Ca^2+^-dependent K^+^ channel (K_Ca_), leading to (8) hyperpolarization and inhibition of odor detection. TGN: trigeminal neuron. AC: adenylyl cyclase; CGRP: calcitonin-related gene peptide; CNG: cyclic nucleotide gated channel; TRPV: transient receptor potential vanilloid; TNFR: TNF-α receptor. TGN: trigeminal nerve ending; OSN: olfactory sensory neuron.

## Data Availability

No new data were created or analyzed in this study.

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
