# Peer review of "Receptors Involved in COVID-19-Related Anosmia: An Update on the Pathophysiology and the Mechanistic Aspects"

_ijms, 2024, doi:10.3390/ijms25158527_

Round 1
Reviewer 1 Report
Comments and Suggestions for Authors
The Manuscript (ijms-3099260) entitled “Receptors involved in COVID-19-related anosmia: An update on the pathophysiology and the mechanistic aspects” aims to evaluate the role of different receptors mediating the impairment in olfactory signal transduction pathways such as ACE2, TMPRSS2 protease, neuropilin 1 (NRP1), basigin (CD147), olfactory, TRPV1, purinergic, and interferon gamma receptors. The complex mechanism that mediates the pathophysiology of the olfactory deficit may reflex different approaches in the anosmia treatment induced by SARS-CoV-2 infection.
This review evaluated a very interesting topic by in depth analysis on the physiological mechanisms of olfactory deficiency. The Manuscript, Figure, and Table are well organized and clear to read, consequently minor revisions are required.
Specific comments:
In the abstract section, please explain all acronyms to better clarify all receptors during the first use. Probably, it should be better to use a list of acronyms in the Manuscript.
In the Introduction, authors should highlight the novelty of this review considering previous studies.
Moreover, in the Introduction authors, among olfactory dysfunction in SARS-CoV-2 infection, should consider also the qualitative deficits (phantosmia) as previously indicated (Ercoli et al., DOI 10.1007/s10072-021-05611-6). The qualitative deficits (phantosmia and parosmia) have been described as a long COVID-19 sequalae.
Reviewer 2 Report
Comments and Suggestions for Authors
This review describes mechanisms of olfactory and gustatory dysfunction in COVID with a focus on receptors and signaling pathways. The review is clearly organized and generally well written. There are some aspects of the review that need substantive improvement. In the introduction, the authors should make use of comprehensive meta-analyses rather than quoting arbitrarily selected studies from the extensive literature. In the section on the olfactory epithelium, the Bowman gland cells deserve more attention. The section on Neuropilin is outdated (the thinking about neurotropism of SARS-CoV-2 has changed) and should incorporate insights gained from the omicron variant. For the section on olfactory (odorant) receptors, the timing of downregulation relative to the onset of anosmia should be considered. The section on interleukins (IL6) needs updates, since the work by Cazzolla et al could not be replicated by many subsequent studies. Some of the sections would benefit from a brief summarizing sentence at the end of the section. Several references need updates or corrections, and some pertinent references should be added. I compile my suggestions separately for minor edits (mostly for style and easier reading).
Specific Points
1. Last sentence in first paragraph of the Introduction: The prevalence of olfactory dysfunction with omicron is quoted as 5.8% to 16.7%. However, a much larger meta-analysis showed a range of 1.9% to 11.7% (von Bartheld and Wang, 2023, Cells 12(3):430). The larger meta-analysis is more likely to reflect the true numbers and should be used.
2. Likewise, in the first sentence of the second paragraph of the Introduction, the authors state that the incidence of anosmia or hyposmia is 3-20%, based on a study of about 1200 subjects. A more recent meta-analysis of over 175,000 subjects reported that percentage at close to 30% of the normal population (Desiato et al., 2021, Am J Rhinol Allergy. Mar;35(2):195-205.). The larger meta-analysis should be cited in this context.
3. In the 5th paragraph of the Introduction, the authors discuss and refer to several small studies on the prevalence of persistent olfactory dysfunction in COVID. Again, the most comprehensive meta-analysis on this topic (Tan et al., 2022, BMJ Jul 27;378:e069503.) should be cited rather than an arbitrary selection of smaller studies.
4. Last paragraph of the Introduction: Many authors disagree with the notion that SARS-CoV-2 induces damage to the olfactory epithelium that is similar to that induced by other viruses, mostly because of the fulminant infection and near complete elimination of sustentacular cells which appears to be rather unique to SARS-CoV-2 (see, e.g., reference #21).
5. In the last paragraph of the section 3., the authors briefly mention Bowman gland cells and some of their functions. In this context, the potential significance of these glands should be expanded. It should be said that these cells, just like the sustentacular cells, express ACE2 and TMPRSS2 and become heavily infected with SARS-CoV-2. Since these glands are innervated by the ACE2-expressing nervus terminalis, there is a potential for a direct neuro-invasion of the brain through this cranial nerve which bypasses the olfactory nerve and olfactory glomeruli. Since the gland cells may provide glucose that powers the energy demands of olfactory cilia, elimination of the gland cells may directly impair olfaction.
6. In the first paragraph of the section 5. “Neuropilin …”, the authors say that SARS-CoV-2 can "use olfactory neurons … to travel from the periphery into the CNS.” This is highly controversial and not supported by more recent evidence. See, e.g., Butowt et al., 2021 Acta Neuropathol Jun;141(6):809-822 or Acta Neuropathol. 2024 Jan 6;147(1):10.; Khan et al. 2021 Cell Nov 24;184(24):5932-5949.e15; Meinhardt et al., 2024 Nat Rev Neurosci Jan;25(1):30-42. There are alternative pathways how SARS-CoV-2 (or its particles) may reach the brain. This controversy should be acknowledged. Since olfactory neurons do not become infected, or only extremely rarely, the neuropilin pathway does not seem to be significant. Also, this is relevant for the last paragraph of this section where the authors say that “SARS-CoV-2 was located in the olfactory neurons and other brain parts.” Note that the evidence is extremely sparse and likely consists of virus particles rather than replicating virus – see Meinhardt et al., 2024. In the context of which receptors facilitate SARS-CoV-2 infection, it should also be considered that insights from the omicron variant are relevant: Omicron still binds ACE2, but it evolved to use cathepsin instead of TMPRSS2, and enters the host cell via endocytosis rather than surface membrane fusion (see Reference #21). This appears to greatly diminish virus entry into olfactory support cells and thus infection in the olfactory epithelium, and therefore largely spares olfaction. Accordingly, the neuropilin pathways is much less relevant than thought initially. Neuropilin appears to be a solution for a problem that does not really exist.
7. In section 6. “Olfactory Receptor …”, the authors say in the 2nd paragraph that downregulation of olfactory (odorant) receptor proteins was suggested to cause anosmia in COVID. In this context, it should be considered that the downregulation of these proteins actually occurs AFTER the onset of anosmia, and therefore cannot be the proximate cause (see References #17 and #24).
8. The sections 6, 7, 8 and 9 would benefit from a brief concluding statement that summarizes the main message developed in that section.
9. In the 9th paragraph of the section 8. “Purinergic …”, the authors should cite the review of Liang, 2020 (Genes (Basel) Apr 30;11(5):493) in the context of intercellular communication between olfactory neurons and supporting cells.
10. In the last paragraph of the section 9. “Interferon …”, the authors say that there was a significant correlation between IL-6 levels and the type of chemosensory dysfunction. Unfortunately, several studies that attempted to replicate this finding failed to support that correlation. Some of these data were summarized in the review by Chen and Wang (2023), the reference #3. Accordingly, the paragraph on IL-6 needs to be rewritten to acknowledge that correlations between IL-6 and chemosensory dysfunction are inconsistent and controversial.
1Minor edits (Note: it would be very useful to have line numbers for the text):
1. Last paragraph in Section 1.: “attracted huge attention …” I suggest adding “and prompted urgency”
2. Same paragraph: “which is similar to that induced by other viruses …” Many authors disagree, because SARS-CoV-2 has such unprecedented, unique effects.
3. Last paragraph of Section 2.: I suggest to use current tense throughout: reviewed – review; we examined – we examine; and connected – and connect; We brought – We bring. Also: “in order to conceive the smell” – should read “in order to perceive the smell”
4. First paragraph of 3. Section: each OSN extend – each OSN extends; that form – that forms; that project to make – that project to form
5. Second paragraph of 3. Section: together make the brush border – together form the brush border
6. Last paragraph of 3. Section: More cell types – Additional cell types
7. Second paragraph of 4. Section: ACE2 receptors expression are … - ACE2 receptor expression is … Also: the information about ACE2 gene expression in lungs, testis and bronchi and lung parenchyma, as well as heart, kidney and gastrointestinal tract appears irrelevant in this context (anosmia). I suggest to shorten.
8. Section 6 and also some paragraphs in Sections 7-9: Because the term “olfactory receptor neuron” is often used in the literature, readers may be confused when you use the term “olfactory receptor” for the odorant receptor protein. I suggest replacing “olfactory receptor” with “odorant receptor” – then there is no ambiguity. This affects the title of Section. 6, the first paragraph of that section, the second paragraph, the last paragraph of Section 7., the 6th paragraph in Section 8 (twice), and the first paragraph of Section 9.
9. Title of Section 6.: I suggest to replace “may be responsible for olfactory dysfunction” with “may contribute to olfactory dysfunction”
10. Section 6., 3rd paragraph: peripherals – periphery
11. Section 8., 5th paragraph: “Anosmia is related, in part, to defect in …” I recommend revising this statement to say “Persistent anosmia appears to be related, in part, to a defect in …”
12. In the 9th paragraph of Section 8., after “intercellular communication.35,” I suggest to also cite the review by Liang, 2020 (Genes (Basel) Apr 30;11(5):493). It seems highly relevant and supportive in this context.
13. Section 9., 2nd paragraph: stem cells mediated neurogenesis – better: stem cell-mediated neurogenesis
14. Section 10., 4th paragraph: “significant overlapping between” – better: Significant overlap between
15. Section 11., 2nd paragraph: with significant sequences … - better: with significant consequences …
16. Next paragraph: in olfactory epithelium – in the olfactory epithelium
17. Next paragraph: odorous molecule – odorant molecule
18. Same paragraph: the intracellular face – the intracellular side or aspect (?)
19. Same paragraph: crosses cribriform bone – crosses the cribriform bone
20. Next paragraph: “As much as positive charge entering through …” – Better: “As long as positive charges enter through …”
21. Section 12., 1st paragraph: “cells seemed to be the starting site for olfaction dysfunction” – better: “cells seem to be the starting site for olfactory dysfunction”
22. Same paragraph: “help in the enhanced CGRP synthesis and release, a process that seemed …” Better: “help to enhance CGRP synthesis and release, a process that seems …”
23. Next paragraph: “by which anosmia can be therapeutically managed.” – that seems a bit too strong. Better: “by which anosmia can be targeted to develop therapeutic approaches.”
24. References: #21 does not give the page numbers 75-90. #34 lists the title of the book, but, according to PubMed, the book series is “Results Probl Cell Differ” and the volume is 52. #41: for consistency, the page numbers should not be in parentheses. #58 is listed as a BioRxiv preprint. This paper underwent significant revisions – relevant in the current context – before the final version was published in Science. Note that the statements from the preprint about virus in olfactory neurons was softened considerably in the final version. #67 is also listed as a BioRxiv preprint. This paper was published in Sci Transl Med. #75: the Journal abbreviation is Eur J Neurosci. #86: The journal abbreviation should be J Neurosci, not J Neurosc. #99: Cazzolla et al – their results could not be confirmed. You could simply refer to the discussion of this topic in Reference #3.
Comments on the Quality of English Language
English language is excellent. I found only a few phrases that would benefit from minor edits, as suggested in the Comments.
